# From Reductionistic Approach to Systems Immunology Approach for the Understanding of Tumor Microenvironment

**DOI:** 10.3390/ijms241512086

**Published:** 2023-07-28

**Authors:** Nicholas Koelsch, Masoud H. Manjili

**Affiliations:** 1Department of Microbiology & Immunology, Virginia Commonwealth University School of Medicine, Richmond, VA 23298, USA; koelschnj@vcu.edu; 2VCU Massey Cancer Center, 401 College Street, Boc 980035, Richmond, VA 23298, USA

**Keywords:** systems immunology, hepatocellular carcinoma, inflammation, immunological pattern

## Abstract

The tumor microenvironment (TME) is a complex and dynamic ecosystem that includes a variety of immune cells mutually interacting with tumor cells, structural/stromal cells, and each other. The immune cells in the TME can have dual functions as pro-tumorigenic and anti-tumorigenic. To understand such paradoxical functions, the reductionistic approach classifies the immune cells into pro- and anti-tumor cells and suggests the therapeutic blockade of the pro-tumor and induction of the anti-tumor immune cells. This strategy has proven to be partially effective in prolonging patients’ survival only in a fraction of patients without offering a cancer cure. Recent advances in multi-omics allow taking systems immunology approach. This essay discusses how a systems immunology approach could revolutionize our understanding of the TME by suggesting that internetwork interactions of the immune cell types create distinct collective functions independent of the function of each cellular constituent. Such collective function can be understood by the discovery of the immunological patterns in the TME and may be modulated as a therapeutic means for immunotherapy of cancer.

## 1. From a Reductionistic Approach to a Systems Immunology Approach

The tumor microenvironment (TME) is a complex and dynamic system that plays a critical role in cancer development and progression. It consists of a variety of cell types, including cancer cells, immune cells, and stromal cells (fibroblasts and endothelial cells), as well as extracellular matrix components and signaling molecules. Solid tumors are generally considered to be more complex than blood cancers in terms of their TME.

Reductionistic approaches to the understanding of the TME involve breaking down the complex system into its individual components and studying them in isolation. This approach has been useful in identifying key molecular pathways and cellular interactions that drive cancer development and progression. However, it has also been criticized for oversimplifying the complex interactions within the TME and ignoring the dynamic nature of the system. In fact, it often fails to capture the full complexity of the TME. For example, studies that focus solely on cancer cells or specific immune cell types may miss important autocrine and paracrine multi-directional mutual interactions, which dynamically change cells of the TME. Another limitation of reductionistic approaches is that they can lead to a narrow focus on specific molecular targets or pathways, which may not fully capture the complexity of the system. For example, targeting a single immune checkpoint molecule may have limited efficacy if other components of the immune system are also contributing to tumor growth and progression. Finally, reductionistic approaches assume that the complex system of interactions in the TME can be understood by assessing the individual function of each cell type present through their cause–effect signaling outcomes. This has been shown by studies evaluating immunosuppressive cell types such as Tregs and MDSCs, which have been identified as one source hindering the effectivity of immune checkpoint therapies [1,2], yet other studies have found the presence of Tregs for modulating inflammation through co-stimulatory molecule inhibition [3]. The depletion of suppressive cell types or pathways may transiently restore the effector functionality of CD8^+^ T cells [4], whereas the efficacy of immune checkpoint therapies targeting T cells is still lackluster, which is particularly evident by RNA-seq methods discovering distinct expression states of immune cells throughout the course of tumorigenesis [5]. Thus, a major issue with research conducted with highly targeted approaches is the deficiency of dynamic assessments by snapshot studies at a single time point, as well as the evaluation of only a single cell type or signaling mechanism in an isolated state. Next-generation sequencing techniques focused on characterizing the TME of solid tumors have exemplified this, as they were unable to elucidate the transformation and generation of cancer [6]. To this end, other methods like live cell imaging can be utilized for the identification of dynamic features in real time. Even though they do not extract nearly as much information as snapshot studies using novel sequencing methods, transitions between phenotypes occurs in temporal fashions [7]. Temporal changes such as these are also seen in cytokine expression, which can further be influenced by dynamic changes in the resident microbial species at the same time during tumor progression [8].

An important finding from recent single-cell sequencing methods is the vast pools of data showing unique transcriptional states of individual cell types. Especially in regard to the canonical understanding of cellular subsets such as macrophages being termed M1 or M2, in which transcriptional data have shown that this classification scheme does not provide the best results for macrophages found in the TME [9,10], which is actually an oversimplification of their heterogeneity [11]. In fact, tumor-associated macrophages (TAMs) identified with high-throughput sequencing methods have been shown to have a much more complex phenotype than the classic dichotomy of M1 and M2, along with other immune cells such as T cells and dendritic cells exhibiting distinct gene expression states [12]. This has also been seen in cancer patients where TAMs that were considered M1 or M2 shared features of one another, such as M2 TAMs producing TNF-α [13,14], as well as macrophages not abiding by the prototypical polarization patterns [15]. This would suggest dynamic cellular states in the TME, which cannot be understood by classic cellular classifications. This is a key point because utilization of standard markers for these cell types may bias analyses within the TME that could be more comprehensively evaluated through scRNA-seq approaches, appreciating the plasticity of the cell subsets, along with the vast amount of transcriptomic data obtained in this approach to decipher their true phenotype. Another issue that stems from approaches reminiscent of reductionistic principles is their intrinsic convolution of the literature with contradictory findings, such as these reports of macrophages enacting functions differing from our canonical understanding, as well as similar findings about unique populations of CD4^+^ and CD8^+^ T cells [16,17,18], and even Tregs producing IFN-γ after immune checkpoint therapy in glioblastoma [19]. However, the use of sequencing techniques generating big data has put us in a position where there is too much information to handle, in which there is a growing appreciation for the inadequacies of targeted studies to fully comprehend the diverse networks of signaling interactions between cell types in the microenvironment [20,21]. Therefore, the biological function and outcome of disease are multifaceted and too complex to understand through rather reductionistic perspectives, as the interactions between cells are constitutively changing in a dynamic manner. Some of our previous works have evaluated the immune system in a way to identify the proportion of immune cells interacting and how it predicts tumor development based on the pattern of immune cells [22,23,24]. Further, our recent work identified that distinct immune patterns could manifest distinct collective function manifested as a complete system, which could be understood through dominant–subdominant patterns of immune cells, along with a mechanistic insight into the active signaling networks within the patterns [25]. Methods such as these are beginning to scratch the surface of understanding the cellular interactions in the TME, which may provide unprecedented insight into how to unify our understanding of cancer, assess it through a holistic lens to view the complex array of cells interacting and devise novel therapeutics to modulate the immunological pattern at superior and inferior levels to generate a curative treatment for cancer patients.

## 2. Application of Systems Immunology Approach from Descriptive to Mechanistic Understanding of the TME

The immune response against cancer functions as a system where immune cells are connected to each other through a mutually interacting network that would be dynamically changing alongside the tumor [26]. Therefore, the outcome of immune responses encompassing the effector and suppressor cells dynamically interacting would depend on the collective immune function. This could only be understood by taking a systems immunology approach that has not been commonly used in the past, but with the advent of novel “omic” technologies, it has begun to induce a resurgence in more holistic research [27]. Systems immunology is becoming more tangible and achievable in its application to diseases like cancer and immune intolerance as a way to make sense of the complex interactions between immune cells and the development of novel therapeutics [28,29]. To this end, recent events have shown the capability of system-level approaches to understand immune responses in COVID-19 patients and discover molecules involved in disease pathogenesis [30,31]. In addition, big data has started to be used as a tool to comprehend the TME alongside biomarker identification, drug discovery, and molecular diagnostics [20,32,33]. Therefore, our ability to understand the TME has increased greatly with sequencing methodologies and big data, exemplifying the utility of such approaches for optimizing treatments by understanding the immune system and its interacting components during immune responses in disease pathogenesis [34]. Regardless, the lack of methods to efficiently comprehend high-throughput sequencing data has resulted in the generation of an immense amount of descriptive knowledge without information regarding the causative mechanisms of disease. Such descriptive information requires a method for shedding light on the mechanism of diseases. To this end, we have recently proposed that identification of the dominant–subdominant relationship in the TME could lead to the discovery of distinct immunological patterns with distinct collective functions as mechanisms by which tumor promotion or inhibition occurs [23]. Such immunological patterns were evident as multilayered from dominant to inferior patterns working together to shape a collective function [23].

The use of “multilayered omics” is a critical factor in our ability to discover immunological patterns and their collective function in the TME. This has already been utilized to stratify cancer patients into groups for the most efficacious therapy regimen for their distinct microenvironment based on characteristics such as metabolism and signaling networks [35,36]. Advances in other “omic” techniques, such as proteomics via mass spectrometry and integrated technological methods, have also been able to comprehend communication networks and mechanisms of processes like host defense and tissue homeostasis [37]. Additionally, “omic” methods focused on understanding the tissue microbiome can provide insights as well, given the role of microbial species in both homeostatic or inflammatory events during dysbiosis [38,39], exemplifying yet another method that can be used to understand the TME at the systems level. Systems-level approaches certainly seem daunting, given the immense amount of information and knowledge required to fully understand the TME, but many of these novel “omic” techniques, like microbiome composition, represent avenues to further study the microenvironment. Although, our ability to make sense of the TME can also be hindered by our own intrinsic ability to comprehend all of the cell types present. There have been many reports that the immunological pattern may more comprehensively elucidate the transition between a disease state and cancer [22,23,24]. In conjunction with advances in big data, the optimization and availability of machine learning have become useful in certain complex cancers like HCC [40,41]. In addition to these novel methods, computational modeling has been used for understanding inflammatory diseases like multiple sclerosis to evaluate interactions between tumors and immune cells [42]. Together, these methods enabled us to understand the immune system from this holistic perspective, which has been lost in the field of immunology over time and replaced by targeted approaches that fixate on a single cell type or mechanism. However, these approaches have helped generate an insurmountable amount of immunological insight; this represents an urgent need for change in the field, as pattern discovery and computational modeling have shown their usefulness in areas of neuroscience and physics to comprehend dynamic systems via top-down models [43]. By shifting our view on these critical methods from future perspectives to a tool that warrants integration into active research, the field of immunology could rapidly change and improve therapeutic efficacy greatly. In light of this, we suggest integrating and utilizing these techniques and methodologies into current research, along with more holistic views such as immunological pattern discovery to understand the TME and the proportion of specific cell types interacting with one another to offer new avenues of therapeutic intervention. This could be as simple as current efforts focused on more rational combinatorial approaches for immunotherapy or new strategies by devising novel therapeutics to modulate the immune system at the pattern level to generate the most robust anti-tumor immune response.

## 3. Pattern Discovery Approach to the Understanding of the TME

Pattern discovery methods such as those employed in our previous works [22,23,24] implicate that the TME in solid tumors cannot be fully understood through the canonical understanding of the immune system, like immune tolerance and suppression, because they are too focused on individual cell types and their isolated function, rather than focusing on the collective function of the immune system as a whole. Ironically, these reductionist approaches miss actual mechanisms of tumor progression or inhibition by overlooking the collective immune function and, in fact, miss the forest for the trees [23]. Other groups have also used pattern recognition through the implementation of novel algorithms to identify immune cell patterns in cancer patients that were associated with different prognoses [44] and responsiveness to immunotherapy [45]. Therefore, pattern discovery methods should be utilized to evaluate the immune system and the network of interactions between the cell types present in the TME, as this can provide a means to modulate the immune patterns present in order to achieve effective anti-tumor immunity. Strategies to modulate immune patterns require more comprehensive data on the TME and are highly dependent on the perspective and level at which the pattern is to be modulated. This has recently been discussed even for the understanding of the pattern of breast cancer dormancy by settling at the mitochondria scale [24] because of its key role in cell death and cell cycle arrest during tumor response to cancer therapies [46]. Interestingly, red blood cells are the only human cells that lack mitochondria, and they do not become malignant. An excessive red blood cell production or polycythemia vera is, in fact, a disorder of the bone marrow or myeloproliferative neoplasm rather than a malignancy of red blood cells [47]. For the understanding of the immunological pattern at the TME, which exists as multilayered patterns, we found that structural cells, fibroblasts in particular, were major coordinators for shaping the immune pattern through influencing signaling networks in the TME [25]. In this same study, we discovered dynamic interactions between fibroblasts and different immune cell types at each stage of disease progression. Each functional signaling pathway showed the structural cells and immune cell type sending and receiving chemokine signals, and dynamically changing depending on the immune pattern and disease status. Moreover, the differential expression of various ligands and their nominal receptors influenced the function of the immune response as a whole within the TME [25]. Thus, such pattern discovery approaches could be used to generate a more comprehensive understanding of the TME and be applied to devising pattern modulatory interventions as cancer immunotherapies. Although this requires an immense amount of work, there are many other components in the microenvironment that need to be studied to evaluate what is the best targetable axis. Exosome signaling is one component that cannot be detected without more specific techniques using microfluidics and biosensors [48,49,50], which could identify additional signaling mechanisms undetectable by single-cell sequencing technologies. In addition, the tissue-resident microbiome is another piece of the TME that can influence the inflammation status and frequency of carcinogenic events [51,52,53,54,55]. Microbial species even fluctuate during the development of cancer, in which these shifts can modulate immune cell phenotypes and facilitate inflammation [56,57], thereby representing another piece of the TME puzzle that can be exploited as a target to influence the overall immune pattern in the microenvironment. Therefore, manipulating the gut microbiome or breast microbiome could also be considered a potential immune-modulating therapy for cancer [58,59]. Interestingly, the underlying microbiome has already been targeted by less intrusive interventions like dietary alterations and supplements [60,61] and also serves as a potential preventative therapy [62]. Lastly, even patients given immunotherapies with suitable responses have detected beneficial microbial species that coincide with elevated immune function [63], further exemplifying the importance of studying the microbiome’s role in the system of immune cells in the TME. Of course, this necessitates additional studies focused on comprehending the key microbial species to be targeted, as well as their byproducts and subsequent effects on the immune pattern. Nevertheless, there are plenty of “omic” technologies that could be used to generate the insight required to devise these novel therapies, which could effectively and non-invasively modulate the immune cells in the TME toward an advantageous functionality.

## 4. Collective Immune Function Emerging from Immune Pattern of Interactions: A Case for Artificial Intelligence

The immune response to the presence of a tumor is induced as a system comprising the innate and adaptive immune cell types interacting with each other and with non-immune cells of the TME to create a collective function that determines the progression or inhibition of cancer. Our recent study using snRNA-seq data was able to show that the cytokines, immune response-related functions, and even metabolic features seen in the collective immune function were unique and independent of the cellular constituents making up the hepatic immune pattern [25]. This was performed by evaluating all of the immune cells together against each of the individual cell types in differential gene expression analyses, highlighting that many functions are only detectable at the immune system level. Few studies attempt to understand the immune system in this manner, especially in regard to the immense amount of interactions occurring between cells in the TME, but characterizing the TME of cancer patients has shown both prognostic value and immunotherapy responsiveness based on the immune cell patterns and their biological function [64]. Assessments of circulating immune cell composition in patients given immunotherapies have also demonstrated dynamic features in peripheral immune cell signatures [65], as well as immune phenotyping of cancers that show different patterns of immune signatures corresponding to patient prognosis [65,66]. This is important in terms of understanding how we can make sense of big data and the immune system, in which patterns of immune cells can be correlated to robust or lackluster immune responses based on the collective function of the immune system generated through mutual cellular signaling events. Further, interactions between immune cells in the TME and tumor cells are becoming more appreciated for their high level of complexity, alongside their influence on plasticity and heterogeneity of both innate and adaptive immune responses [67,68,69]. Even transcriptional signatures of mixed immune cell populations have identified components involved in disease pathogenesis, but more importantly, understanding how multiple signaling stimuli are integrated and influence immune cells in inflammatory microenvironments can provide a more comprehensive understanding of the TME [70]. The collective function of the immune response would also dynamically change in the TME through multiple signaling axes from the immune, structural, and tumor cells present and engaging in mutual signaling interactions. Such multi-directional mutual signaling interactions may change depending on the pattern of immune cells in the TME. To this end, dysregulation of the immune system and the proportion of individual immune cell types, along with their production of functional signaling molecules and the expression of activating or inhibitory receptors and ligands, are critical for understanding the development of cancer [71]. Even epigenetic studies using systems immunology via a pattern discovery approach in the TME have identified patterns of infiltrating immune cells in cancer patients that correlate to DNA methylation regulator genes, in which scoring systems could help predict patient survival [72]. This represents another level of understanding of the immune response, where epigenetic alterations may polarize cells toward a set of functional signaling that perturbs the immune patterns and facilitates the retention of functions fostering tumor survival. These reports help signify the fact that the immune system manifests variable functions when evaluating the individual cell types, but their mutual interactions together as a whole generate a collective immune response that dictates if a tumor will persist or be eliminated. This can be likened to the proportion of ingredients needed in a recipe and how significantly these components impact flavor, texture, and the appearance of a dish. Chefs experiment with the proportion of ingredients to generate new unique flavor combinations, as well as different textures and appearances that would be desirable to the consumer. In other words, the use of different ingredients in varying amounts can result in pleasant and unpleasant flavors in a dish, similar to the pattern of immune cells, where an excess or deficiency of a particular cell type in the immunological pattern, along with TME conditions lead to different outcomes of disease progression (Figure 1).

For the accuracy of the immunological pattern and, in turn, designing effective immunomodulatory therapeutics, we need to take into consideration multilayered mutual interactions at the TME, which is not limited to the receptor–ligand interaction and involves exosomes, microRNAs, tissue microbiome, tissue metabolome, etc. Such a complex biological system requires advanced algorithms to be processed. To this end, advances in artificial intelligence are expected to revolutionize research approaches as well as the interpretation of new observations by decoding the immunologic complex systems of the TME [73]. Very recently, artificial intelligence has been used for breast cancer grading [74,75], depiction of mouse immune microenvironment during breast cancer [76], and breast cancer diagnosis [77,78], to this end. The supervised learning-based support vector machine (SVM) algorithm is one of the widely used methods for the analysis of multi-omics data. They are used for cancer biomarkers for finding cancer subtypes. Random forest (RF) algorithms are also used as cancer classifiers. In addition to supervised learning-based SVM and RF algorithms, unsupervised learning approaches such as autoencoders are used to reduce the size of multi-omics big data [79]. Among these algorithms, the prediction model seems to be more relevant to immunological pattern discovery and detection of collective immune function compared with model-based integration, where the function of individual cells is combined [80]. This is because we have discovered that collective immune function is independent of the function of the cellular constituent and beyond the sum of individual cell function; rather, the collective function is predicted based on dynamic interactions of immune cells [25]. To this end, DeepGO is one of the most successful approaches to predicting protein function [81]. Therefore, algorithms that can predict immune function based on patterns of cellular interactions would be more useful tools for the understanding of TME and the development of novel immunotherapies.

## 5. Immunotherapy of Cancer: Immune Modulation or Immune Cell Induction?

Current immunotherapies are guided by reductionist approaches and offer a specific immune cell type, such as T cells, NK cells, or antibodies, for the treatment of breast cancer. Although decades of experience with these immunotherapeutics have proven to prolong patients’ survival, a cure for advanced-stage disease remains elusive. The review of the literature in this essay suggests that doing more of the same without considering the complexity of the TME would not result in a different outcome in the curative treatment of breast cancer. In fact, breaking down various therapies into a single cell type or signaling pathway like anti-CTLA4 immunotherapy [82], anti-PD-L1 immunotherapy [83], CAR-NK cell therapy [84], and antibody therapies [85] exemplify the shortcomings of this approach, given the fact that none of these therapies could offer a cure to cancer patients. One reason for the transient efficacy of such immunotherapies is that it would take time for an altered immune pattern within the TME to control new anti-tumor T cells by forcing them to adapt to the TME. Additional efforts in this reductionist route are expected not to go beyond incremental advances in cancer immunotherapy. Therefore, immune modulatory interventions that may change the immune patterns could serve as a better avenue for therapeutic development, as so many immunotherapies targeting a single cell type solely focus on augmenting individual immune cell responses. For example, CAR T cell therapies have been quite promising for hematological cancers because of the less complexity of the TME, but they are still appraised as rather exploratory and require much more research, despite repeated failures to become effective for solid tumors, perhaps because of the complexity of the TME [86,87]. This makes CAR T cell therapy ineffective against breast cancer [88]. On the other hand, advances in systems immunology could result in breakthroughs in understanding the TME and, in turn, the development of immune modulatory approaches for targeting and fixing an impaired pattern of immune cell interactions or their collective immune function [89]. Artificial intelligence has been applied to the prediction of responses to immunotherapy based on immune signatures as well as neoantigen prediction based on the MHC background of patients for designing personalized cancer vaccines. Future application of artificial intelligence in systems immunology is expected to be focused on the discovery and modulation of immunological patterns as a therapeutic means for breast cancer patients. It is also expected to revolutionize our understanding of the immune system beyond cancer as it comes to the participation of the immune response in the physiological function of the organs as well as defending the host from pathogens. Major differences between reductionistic and systems immunology approaches in the understanding of the TME are summarized in Table 1.

## Figures and Tables

**Figure 1 ijms-24-12086-f001:**
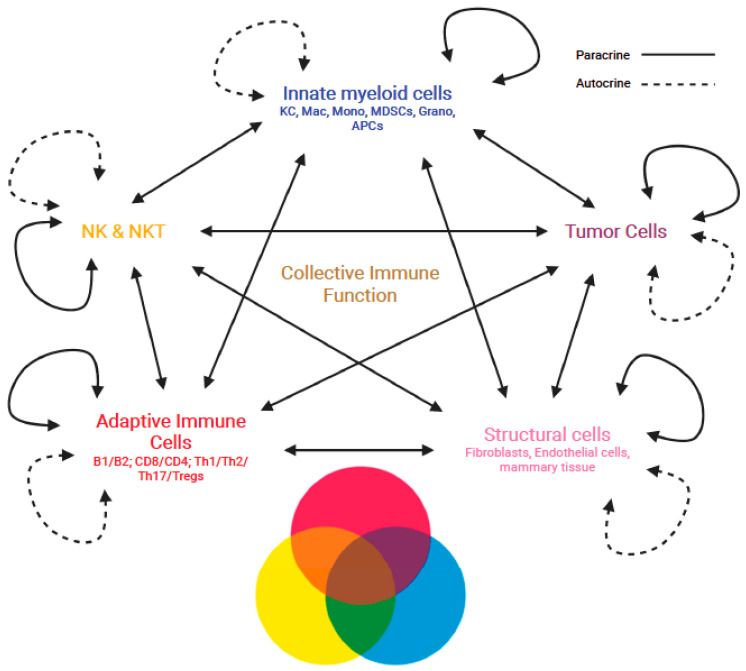
Manifestation of emergent collective immune functions through the multi-directional mutual interactions of immune cells in the TME. Multi-directional mutual interactions of the immune cells and non-immune cells in autocrine and paracrine fashions dynamically shaping the collective immune function in the TME.

**Table 1 ijms-24-12086-t001:** Reductionistic vs. Systems Immunology approach to TME.

	Reductionistic	Systems
**Mechanism**	Cause-Effect	Pattern of mutual interactions
**Function**	Cellular or humoral immunity	Emergent collective immune function
**Outcome**	Effector vs. Suppressor cells	Dominant-subdominant interactions
**Immunotherapy**	Inducing killer immune cells and inhibiting suppressor cells	Modulation of the immune pattern in the TME

## Data Availability

Not applicable.

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
