# Peer review of "From Reductionistic Approach to Systems Immunology Approach for the Understanding of Tumor Microenvironment"

_ijms, 2023, doi:10.3390/ijms241512086_

Round 1

Reviewer 1 Report

Revision of the review: “From reductionistic approach to systems immunology approach for the understanding of tumor microenvironment”.

The review describes the current view of the tumor microenvironment (TME), which is a simplified view/approach, which is quite limiting considering the complexity of it and proposes to use systems immunology approach to further understand the complexity of TME. These approach would help to improve and/or to developed new immunotherapies against cancer.

I consider that the article is of interest for the scientific community however I have major concerns that lead me to propose its rejection, at least for now.

I proposed to post-pone the publication of the review after the acceptance/publication of the paper “Koelsch et al, submitted”. The reason such decision is that major claims in the review are mainly based in the discovery done in “Koelsch et al, submitted”. However, considering that “Koelsch et al, submitted” that has not been accepted for publication or published and it is not sure that it is going to be, such reference cannot be used in the review, as we cannot confirm the veracity of the results. Thus, considering that this review will be published before the paper under submission. I strongly encourage that only after publication of the “Koelsch et al. submitted”, the review should be then be re-submitted and reviewed.

Another important point is that considering that it is review, it lacks images to help the reader to follow it. It is very exhaustive in its reading and more figures are required.

Good Enghish quality

Author Response

 I proposed to postpone the publication of the review after the acceptance/publication of the paper “Koelsch et al, submitted”. The reason such decision is that major claims in the review are mainly based in the discovery done in “Koelsch et al, submitted”. However, considering that “Koelsch et al, submitted” that has not been accepted for publication or published and it is not sure that it is going to be, such reference cannot be used in the review, as we cannot confirm the veracity of the results. Thus, considering that this review will be published before the paper under submission. I strongly encourage that only after publication of the “Koelsch et al. submitted”, the review should be then be re-submitted and reviewed.

 Response: We have updated the status of our unpublished paper to in-press, as it was recently accepted for publication in Scientific Reports..

Another important point is that considering that it is review, it lacks images to help the reader to follow it. It is very exhaustive in its reading and more figures are required.

Response: We added Table 1.

Reviewer 2 Report

The present study investigated the different approaches and mechanisms for understanding the tumor microenvironment. The manuscript is well-written, and the approaches and mechanisms have been described in detail. It was a great pleasure reading it. However, only one issue needs addressing. 

  1. Hepatocellular carcinoma and inflammation keywords are not mentioned in the abstract. Emphasis should be placed on these keywords in the abstract.

Author Response

Hepatocellular carcinoma and inflammation keywords are not mentioned in the abstract. Emphasis should be placed on these keywords in the abstract.

Response: The manuscript is focused on the pattern of the immune responses in the TME rather than on inflammation in HCC. We clarified this throughout the manuscript.

Round 2

Reviewer 1 Report

I consider that the paper can be published, since one of my major concerns was addressed. The paper mentioned in the review: "Koelsch et al, Sci Reports, in press" will be published. I still would recommend to delay the publication of the review for after the publication of the paper of the authors.